# Structure and Optical Properties of Co-Sputtered Amorphous Silicon Tin Alloy Films for NIR-II Region Sensor

**DOI:** 10.3390/ma12244076

**Published:** 2019-12-06

**Authors:** Xiang-Dong Jiang, Ming-Cheng Li, Rui-Kang Guo, Ji-Min Wang

**Affiliations:** School of Optoelectronic Science and Engineering, University of Electronic Science and Technology of China, Chengdu 610054, Chinajmwang@uestc.edu.cn (J.-M.W.)

**Keywords:** a-SiSn alloy thin films, a-Si network, structure, nanocrystals, near-infrared brain imaging

## Abstract

Near-infrared brain imaging technology has great potential as a non-invasive, real-time inspection technique. Silicon-tin (SiSn) alloy films could be a promising material for near-infrared brain detectors. This study mainly reports on the structure of amorphous silicon tin alloy thin films by Raman spectroscopy to investigate the influence of doped-Sn on an a-Si network. The variations in TO peak caused by the increase in Sn concentration indicate a decrease in the short-range order of the a-Si network. A model has been proposed to successfully explain the non-linear variation in Raman parameters of I_TA_/I_TO_ and I_LA+LO_/I_TO_. The variations of Raman parameters of the films with a higher deposition temperature indicate the presence of SiSn nanocrystals, though the SiSn nanocrystals present no Raman peaks in Raman spectra. XRD and TEM analysis further illustrate the existence of nanocrystals. The ratio of photo/dark conductivity and optical bandgap results demonstrate that the films can be selected as a sensitive layer material for NIR-II region sensors.

## 1. Introduction

Si is the most widely used semiconductor material in integrated optical devices. The use of group IV binary or ternary alloys with Sn and Ge (SiGe [1], SiSn [2,3,4], SiGeSn [5]) is one way to apply Si to silicon-based devices because of their promising possibilities for band gap engineering and structural modification. SiSn received widespread attention due to the energy gap (0.75–0.95 eV) being suitable for optical communication [6]. Several reports have demonstrated that the band gap of SiSn alloys can be adjusted in a wind range by controlling the Sn concentration [7]. The wide band gap variations of the SiSn thin films means that they can be used in the near-infrared (NIR) to far-infrared region [8]. Many scientists so far have focused on highly doped Sn to achieve various goals. For example, the SiSn crystallization temperature can be reduced with a substitutional Sn content of around 30% [2]. In order to make SiSn alloys suitable for optical application, Masato Oda et al. [9] utilized interacting quasi-band (IQB) theory and found that the indirect–direct gap crossover in Si_1−x_Sn_x_ occurs around x = 0.67 with Eg = 0.87 eV. There are also many studies on the optical and electrical properties and surface microstructure of SiSn films with an Sn concentration of 10%–51% [10,11,12].

In recent years, much attention has been paid to the medical applications of the infrared band, especially the near-infrared window of 1000–1700 nm (NIR-II). This band has great application value in NIR brain imaging due to its long tissue penetration depths and ultralow tissue backgrounds [13,14]. We attempt to make SiSn films apply in the sensitive layer of near-infrared brain imaging detectors. The variations of the optical bandgap of SiSn films versus Sn concentration are almost a linear relationship [15] and highly doped Sn will cause the absorption band of the SiSn films to move to the mid–far infrared band. In order to locate the absorption band of the SiSn films in the near-infrared region, the SiSn films with a lower Sn concentration (0–10%) was necessary and the variation of the structure, which is crucial for the design and optimization of NIR brain imaging devices, also needed to be investigated in detail. Raman spectroscopy was chosen as a research method because it is a cheap, easy to operate and nondestructive tool [16,17]. The structure of SiSn films with a low Sn concentration, to our best knowledge, has not been reported yet. Here we report the a-Si_1−x_Sn_x_ alloy thin films prepared by RF magnetron sputtering at a relatively low concentration of Sn, and a series of Raman parameters versus Sn concentration was mainly studied. X-ray diffraction (XRD) and transmission electron microscopy (TEM) were used to support the Sn nanocrystallization process that may occur during the deposition process. The ratio of photo/dark conductivity and optical bandgap prove it is suitable for NIR-II region sensor.

## 2. Materials and Methods

The a-Si_1−x_Sn_x_ alloy thin films deposited on a 25 mm × 15 mm × 1 mm Si (100) wafer with a working pressure of 4.3 × 10^−1^ Pa by means of an RF magnetron co-sputtering method. The RF power was fixed at 200 W with a power density of about 2.55 W/cm^2^. Argon (Ar) acts as a working gas with a flow of 75 sccm. The distance between the substrate and the target was 10 cm. Prior to deposition, the Si substrates were ultrasonically cleaned separately in acetone and ethanol for 15 min. The substrates were heated to 150 °C before deposition. The starting materials were metallic tin particles (99.99% pure) placed on the single crystal silicon target (99.999% pure). The x of a-Si_1−x_Sn_x_ alloy thin films varied from 0 to 0.1. A deposition temperature varied from 150 °C to 300 °C. All films discussed in this paper have a deposition time of 30 min.

A RENISHAW inVia Raman Microscope in a backscattering configuration at a wavelength of 514 nm was used to analyze the microscopic molecular structure. Grazing incidence X-ray diffraction patterns were recorded on a X’pert(PANalytical, Almelo, Netherlands) diffractometer using monochromated Cu Kα1 radiation (λ = 0.15418 nm). TEM was recorded on an FEI Tecnai F20 field emission gun microscope. The ratio of photo/dark conductivity was tested with a Keithley resistor. We use a Cary5000 (Agilent, Santa Clara, CA, USA) UV-VIS-NIR spectrophotometer (175–3300 nm) to get the transmittance, and the optical bandgap as calculated with the Tauc equation.

## 3. Results and Discussion

In order to understand the influence of Sn-doped concentrations on lattice vibrations of a-Si_1−x_Sn_x_ alloy thin films, their Raman spectra (Figure 1) were recorded. A typical Raman spectrum of a-Si contains a total of four phonon modes, namely, one longitudinal acoustic (LA), one transverse acoustic (TA), one longitudinal optical (LO) and one transverse optical (TO) branch. The location of TO peak (W_TO_), which is proportional to the tetrahedral network (sp3 hybridization) and sensitive to the short-range order [18], are shifted toward lower wavenumbers in the case of S5 compared to S1 (with the Sn concentration increasing from 0% to 10%). The width of the TO peak (Γ_TO_), as shown in Figure 2b, is broader by 21.26 cm^−1^ with Sn concentration variation. Both the shift of W_TO_ to a low wavenumber and the broadening of the TO peak with increasing concentration of Sn indicate a greater disorder in a short range. The value of ∆θ is increased from 12.21° to 15.72° (Figure 2c) according to the linear formula Γ_TO_ = 15 + 6∆θ. A plausible rationale for this behavior is that both W_TO_ and Γ_TO_ are related to the Si–Si bond-angle variation ∆θ [19], and the doped Sn of a-Si_1−x_Sn_x_ alloy thin films increases the distortion of the a-Si network due to the difference in size between the Sn atoms and Si atoms. The intensity of LA (I_TA_), TA (I_TA_), LO (I_LO_) and TO (I_TO_) are all related to the a-Si network. Zotov et al. [20] have reported that the Raman characteristic parameters I_LA+LO_/I_TO_ and I_TA_/I_TO_ are related to defect and medium range in a-Si thin films, respectively. The increases in I_LA+LO_/I_TO_ and I_TA_/I_TO_ (Figure 2d,e) indicate a generation of more defects and a decrease in medium-range order, respectively.

An interesting phenomenon was observed after further analysis of Figure 2. The three parameters Γ_TO_, W_TO_ and ∆θ related to the TO mode are almost linear with the variations of Sn concentration, while the intensity ratios I_LA+LO_/I_TO_ and I_TA_/I_TO_ are non-linear with the variations of Sn concentration. Referring to the introduction of hydrogen (H) atoms into the a-Si:H thin films, we propose a hypothesis to explain the non-linear relationship. As depicted in Figure 3a, a silicon ring exists locally in amorphous silicon networks and all Si atoms have four bonds. Hence an H atom contains only one valence electron, the entry of H atoms into amorphous silicon grids breaks the Si rings in the a-Si network as shown in Figure 3b. Unlike how H atoms break the Si rings, we believe that the Sn atom will form four SiSn bonds with the surrounding four Si atoms, thus the Si ring can still hold a ring configuration as shown in Figure 3c. Of course, the introduction of the Sn atom will break four Si–Si bonds. This hypothesis could be the reason for the non-linear relationship. The intensity of TO mode I_TO_ is strongly related to the tetrahedral network, so I_TO_ will decrease with the decreasing amount of Si–Si bonds. While the I_LA_ and I_LO_ are associated with the presence of closed silicon rings in the amorphous network, the reduction of I_LA_ and I_LO_ is limited. In general, the effect of the introduction of the Sn atom on I_TO_ is greater than that on I_LA_ and I_LO_, resulting in the non-linear relation.

Another interesting phenomenon can be also found in Figure 2, that I_LA+LO_/I_TO_ and I_TA_/I_TO_ increase sharply in the range of a low Sn concentration but increase slowly in a relatively high Sn concentration. The solubility limitation of Sn atoms in a-Si networks may be responsible for this non-linear relationship. Before the Sn concentration reaches the solubility limitation of Sn atoms in a-Si networks, the intensity of the TO mode decreases sharply, resulting in a rapidly increasing rate of I_LA+LO_/I_TO_ and I_TA_/I_TO_. The concentration of Sn is further increased to reach the solubility limitation, so the concentration of substitutional Sn atoms no longer increases, which suppresses the TO mode rather than the TA, LA and LO modes. Therefore, in the range of a higher Sn concentration, the increase rate of I_LA+LO_/I_TO_ and I_TA_/I_TO_ will also slow down. This non-liner also indicates the formation of the SiSn bonds that were discussed in the previous discussion.

Figure 4 shows the Raman spectra of a-Si_1−x_Sn_x_ alloy thin films at different deposition temperatures with an Sn concentration of 6%. Significant variations are not observed in the Raman spectra of the films at a deposition temperature of room temperature, 150 °C and 200 °C. The Raman spectra of the a-Si_1−x_Sn_x_ alloy thin films do not show any characteristic peaks for crystalline Si, indicating the presence of amorphous Si. The presence of c-Si is first observed at a higher deposition temperature of 300 °C. W_TO_ moves to a wavenumber of 511 cm^−1^, which is a major Raman peak in c-Si.

More details of the a-Si_1−x_Sn_x_ alloy thin films Raman characteristic parameters are given in Figure 5. W_TO_ is slightly shifted toward a higher wavenumber and the reduction in ∆θ and Γ_TO_ both indicate an improvement in the a-Si network. One thing that should be noted is that there is a big variation in W_TO_, Γ_TO_ and ∆θ between a deposition temperature of room temperature and 150 °C, but little variation between deposition temperatures of 150 °C, 200 °C and 300 °C. Therefore, we believe that the amorphous silicon network has been fully ordered at a deposition temperature of 150 °C. This means that the Raman parameters W_TO_, Γ_TO_, ∆θ, I_LA+LO_/I_TO_ and I_TA_/I_TO_ should be relatively stable between 200 °C and 300 °C. However, I_LA+LO_/I_TO_ and I_TA_/I_TO_ continue to decrease. This anomaly contradicts the previous analysis.

We propose a reasonable mechanism to explain this phenomenon. Alexander et al. [21] have reported the presence of SiSn nanocrystals in Si_1−x_Sn_x_ alloy thin films, so we believe that our samples also produce SiSn nanocrystals. Once the deposition process is completed, the total number of Sn atoms in the film is constant. On the one hand, the increase in temperature will promote the growth of SiSn nanocrystals. The formation of SiSn nanocrystals consumes Sn atoms, so the substitutional Sn atoms in the a-Si network are reduced. On the other hand, there is an increase in the amount of Si–Si bonds, so the intensity of the TO mode, which is related to the vibration of Si–Si bond, is increased. Therefore, I_LA+LO_/I_TO_ and I_TA_/I_TO_ maintain a tendency to decrease as temperature increases.

To further confirm the formation of nanocrystals, we use XRD and TEM as an auxiliary method. Figure 6 shows the XRD spectra of a-Si_1−x_Sn_x_ alloy thin films at a deposition temperature of room temperature (RT) and 150 °C with an Sn concentration of 0.06%. No characteristic diffraction peaks are observed in the sample with a deposition temperature of RT. The diffraction peaks of Sn deposition at 150 °C were sharp and intense, indicating their highly crystalline nature. Characteristic diffraction peaks of Si and SiSn alloys were not observed. Actually, there was no significant difference in the XRD patterns of Sn and SiSn films, indicating the presence of amorphous Si [22]; this result can be a proof of our speculation. According to the Scherrer formula, we calculated that the mean size of SiSn nanocrystals is about 10–15 nm. TEM images of a-Si_1−x_Sn_x_ alloy thin films at a deposition temperature of 150 °C, as presented in Figure 7a,b, indicate the coexistence of the amorphous matrix and nanocrystals. It can be clearly seen in Figure 7a that there are nanoparticles dispersed in the films, and these should be SiSn nanocrystals. The size of nanocrystals (about 10 nm) in TEM images is very close to the result of XRD, which could further confirm the size of the SiSn nanocrystals. Next we use Digital Micrograph to calibrate the interplanar crystal spacing and a crystal face (200) of Sn was determined. Through the literature and analysis mentioned above, this could indicate the formation of SiSn nanocrystals. The diffraction pattern, as shown in the inset of Figure 7b, consist of a series of poly-crystal rings belonging to Sn nanocrystals. The diffraction ring is relatively wide, probably because the nanoparticle size is too small. Both the width of the diffraction ring and the TEM images show that the nanocrystals are small, which is consistent with the XRD calculation results. Every single result of Raman, XRD or TEM does not seem to be sufficient to prove the formation of SiSn nanocrystals, but the combination of the three analytical results is considered sufficient to demonstrate the formation of SiSn nanocrystals.

In addition, we also studied the ratio of photo/dark conductivity variation of the a-Si_1−x_Sn_x_ alloy thin films prepared at different Sn concentrations. The ratio of photo/dark conductivity of the films first increases and then decreases as shown in Figure 8. The reason for this is because when the concentration of Sn in the films is at a low level, the Sn atoms replace the Si atoms in the a-Si lattice to increase the bright conductance of the films, resulting in an increase in the ratio of photo/dark conductivity. When the Sn concentration of the films increases to a relatively high value, the Sn atoms cannot replace the Si atoms in the a-Si lattice, thereby providing a large amount of free electrons, resulting in an increase in the dark conductivity of the a-Si_1−x_Sn_x_ alloy thin films. Finally, we got the optical band gap of the SiSn film at different Sn concentrations as shown in Figure 9. The absorption wavelength corresponding to the band gap falls just in the near-infrared II region. We believed that both the ratio of photo/dark conductivity and optical bandgap of a-Si_1−x_Sn_x_ alloy thin films indicate it can be used in a sensitive layer of the NIR-II sensor.

## 4. Conclusions

In conclusion, a comprehensive Raman analysis of a-SiSn alloy thin films was given. By doping a relatively low concentration of Sn, we demonstrate the dependence of an amorphous matrix short-range order on the Sn concentration of alloy thin films. This indicates that the increase of Sn concentration will lower the short-range order of the amorphous matrix. A plausible rationale for the intensity reduction of the TO mode is that the substitutional Sn atoms reduce the amount of Si–Si bond configurations. While the of Si–Si bond decreases with the introduction of Sn atoms, the Sn atom does not break the Si rings. Furthermore, we study the effect of deposition temperature on the Raman spectra of a-SiSn alloy thin films. The short-range order of the amorphous matrix is improved by increasing the deposition temperature. The Sn nanocrystals are observed by the analysis of the Raman characteristic parameters and the XRD characteristic diffraction peaks. The TEM spectra show the Sn nanocrystals are dispersed in the a-SiSn alloy thin films. Our study is important for the application of SiSn materials to near-infrared brain imaging technology. Finally, both the ratio of photo/dark conductivity and optical bandgap of a-Si_1−x_Sn_x_ alloy thin films indicate that the film has the potential to be applied to the near-infrared sensor.

## Figures and Tables

**Figure 1 materials-12-04076-f001:**
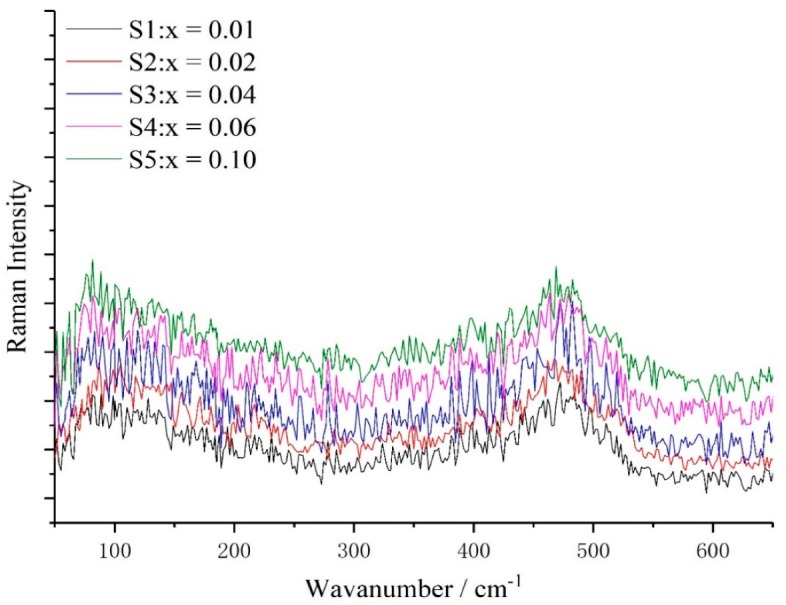
Raman spectra of a-Si_1−x_Sn_x_ alloy thin films with different Sn concentration x.

**Figure 2 materials-12-04076-f002:**
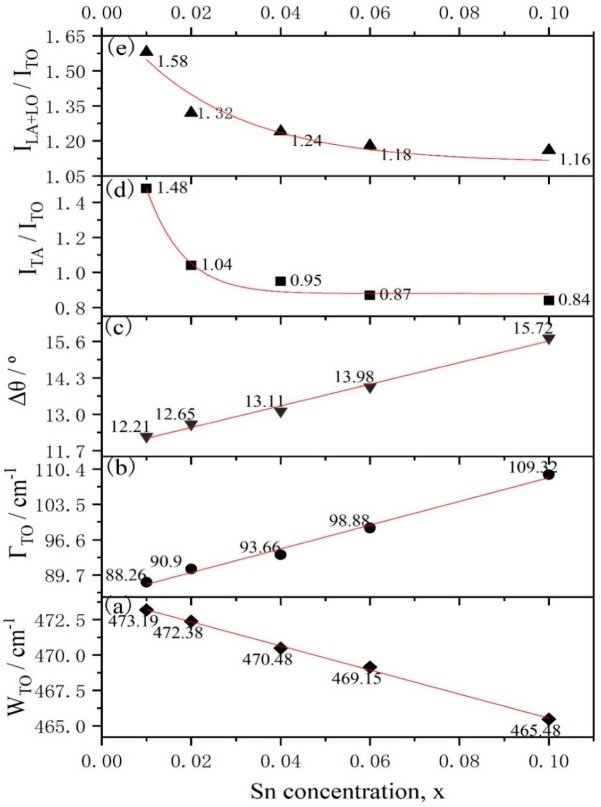
Raman characteristic parameters of a-Si_1−x_Sn_x_ alloy thin films versus different Sn concentration x ((**a**) 0.01, (**b**) 0.02, (**c**) 0.04, (**d**) 0.06, (**e**) 0.1).

**Figure 3 materials-12-04076-f003:**
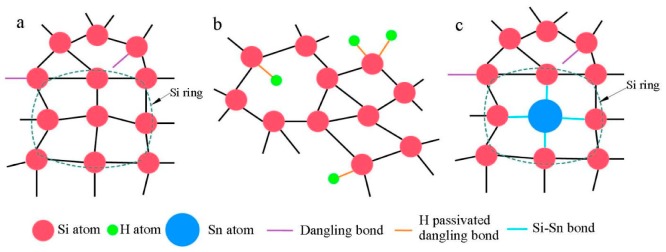
(**a**) Locally existing silicon rings in amorphous silicon structures, (**b**) an amorphous silicon structure after the introduction of an H atom and (**c**) amorphous Silicon structure after the introduction of an Sn atom. The Si ring can be retained after the introduction of Sn.

**Figure 4 materials-12-04076-f004:**
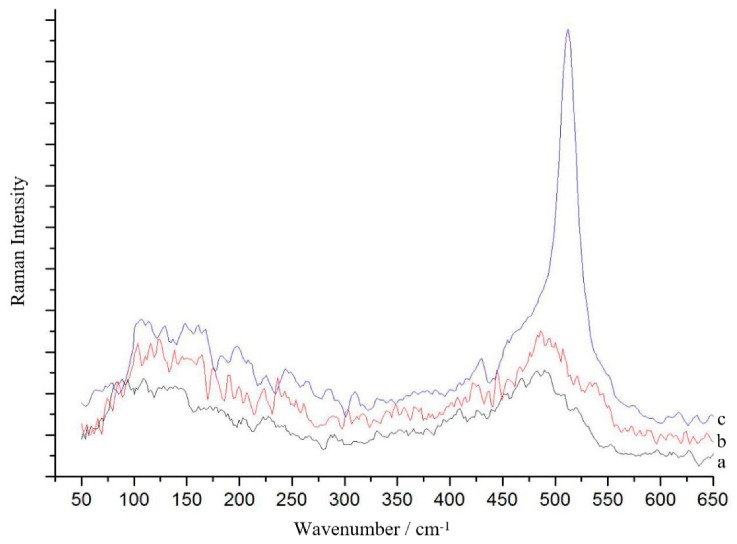
Raman spectra of a-Si_1−x_Sn_x_ alloy thin films grown at (**a**) 150 °C, (**b**) 200 °C, (**c**) 300 °C.

**Figure 5 materials-12-04076-f005:**
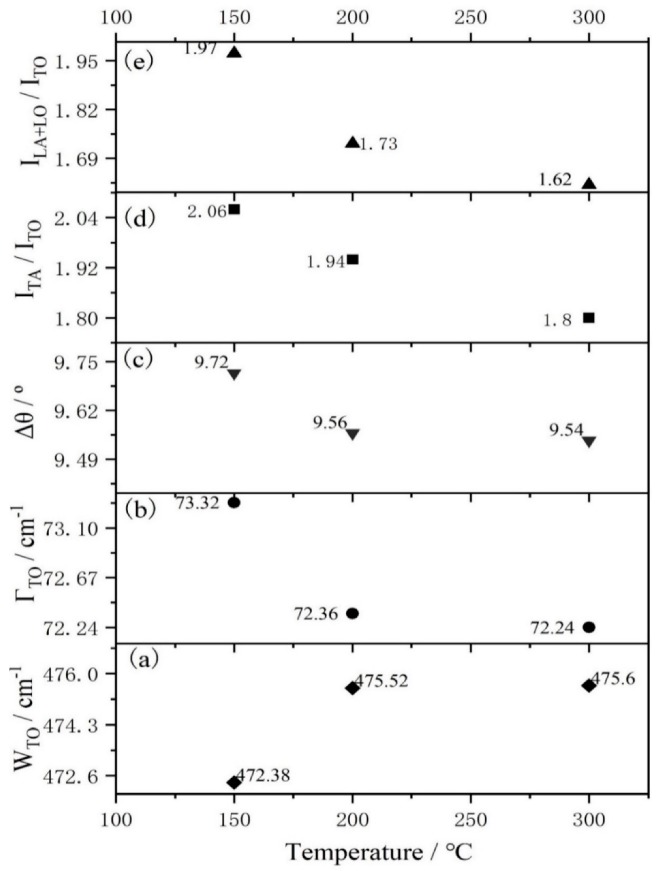
Raman characteristic parameters of a-Si_1−x_Sn_x_ alloy thin films versus grown temperatures of room temperature (RT) and 150 °C.

**Figure 6 materials-12-04076-f006:**
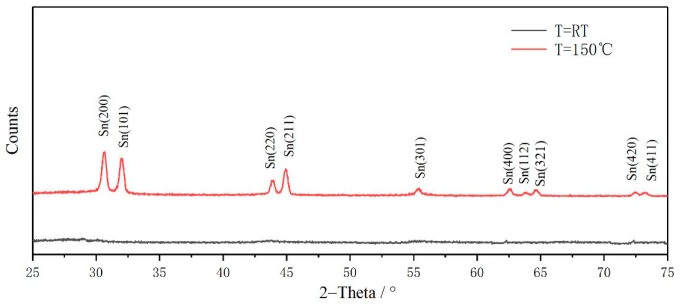
XRD spectra of a-Si_1−x_Sn_x_ (x = 0.06) alloy thin films at a grown temperature of room temperature (RT) and 150 °C.

**Figure 7 materials-12-04076-f007:**
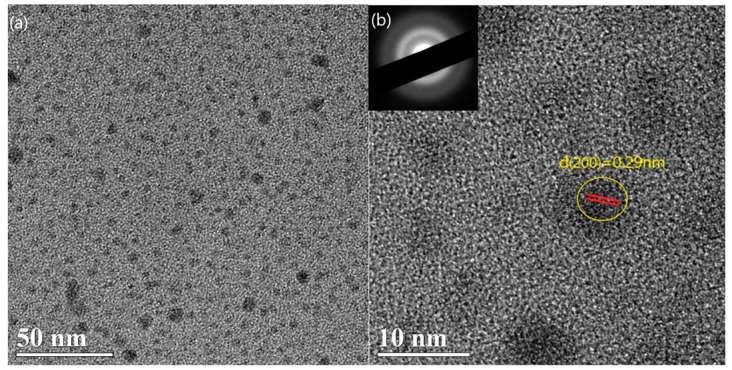
TEM (**a**) and HRTEM (**b**) images of a-Si_1−x_Sn_x_ (x = 0.06) alloy thin films at a grown temperature of 150 °C.

**Figure 8 materials-12-04076-f008:**
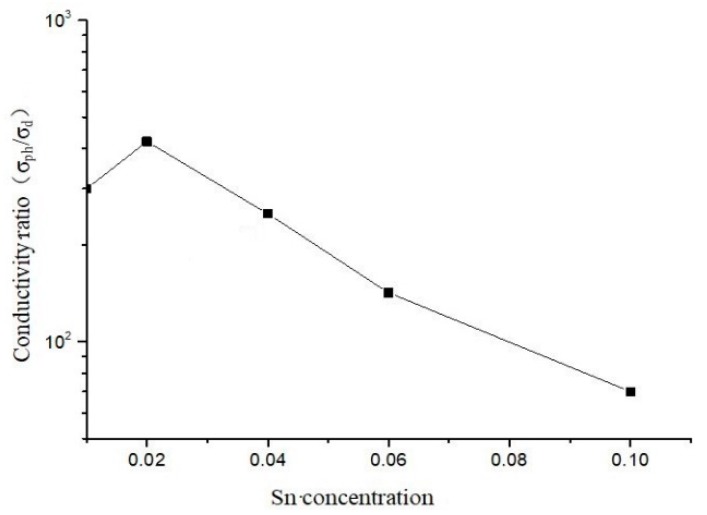
Ratio of photo/dark conductivity of a-Si_1−x_Sn_x_ alloy thin films.

**Figure 9 materials-12-04076-f009:**
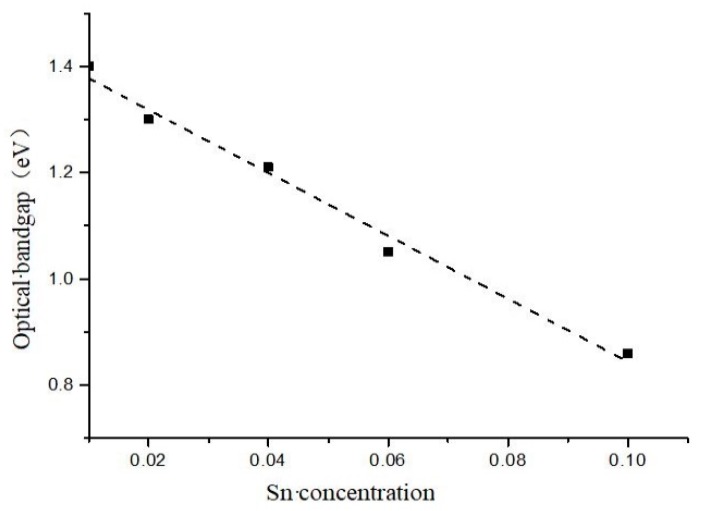
Optical band gap of a-Si_1−x_Sn_x_ alloy thin films for various values of x.

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
