# Peer review of "Structure and Optical Properties of Co-Sputtered Amorphous Silicon Tin Alloy Films for NIR-II Region Sensor"

_materials, 2019, doi:10.3390/ma12244076_

Round 1
Reviewer 1 Report
According to manuscript entitled „Structure and optical properties of co-sputtered amorphous silicon tin alloy films for NIR-â…¡ region sensor“ by authors X. D. Jiang, M. Ch. Li, R. K. Guo, J. M. Wang.
The manuscript presents magnetron sputtering deposition, structure and properties of amorphous SiSn alloy films for infrared sensor. The results and discussion are well presented.
The manuscript is worth to be published after minor corrections.
Describe how the authors have been calculated the optical band gap. In the Materials and Methods section, the authors stated „...optical bandgap were tested by ... and ultraviolet spectrophotometer, respectively.“ This is not correct. My guess is that the UV-VIS-NIR spectrophotometer is used to measure the reflectance spectra and based on these spectrophotometric data, the optical band gap is determine. The authors must descripe the spectrophotometer and the spectral range used as well as the calculaion method.
Author Response
Thank you for your thoughtful review, we have carefully taken your comments into consideration. Here's the response for your question. Please see the attachment.

Reviewer 2 Report
The paper is very interesting, the analysis is very well written and the results are supported by the data. Therefore I recommend the publication of the paper in its present form.
Author Response
Thanks for your careful review!
Reviewer 3 Report
Authors present some results of structures and optical characterization of amorphous silicon tin alloy films for NIR-â…¡ region produced by co-sputtering. An attempt to understand the results is made. The use of a-SiSn alloy thin films on NIR sensor proved to have potential and deserves further research. The results of the characterization of the SiSn films with low Sn concentrations gave some indications but should be furhter extended either in what concerns Raman and TEM. The raman spectra and the adjacent contradictions on measurements presented at figure 2 are mentionned (page 4) but not sufficiently explained. The explanation at page 5 and next requires further proof. Statements on line 158-160 (page 6) needs reinforcement. I suggest further measures to be performed. The conclusion presented are not enough justified and therefore can not be accept. I suggest the authors to further work on the issues raised in this work and resubmit the paper after solidifying the conclusions.
Author Response
We want to begin by thanking you for writing your constructive suggertion, the attachment is our detailed modification for your point.

Round 2
Reviewer 3 Report
Authors improved the paper. However further research will certainely increase the significance and impact of the paper.